# Relation Between COVID-19 Infection and Vaccine and Menstrual Cycle Changes of Portuguese Adolescents in Higher Education

**DOI:** 10.3390/healthcare13010002

**Published:** 2024-12-24

**Authors:** Zélia Caçador Anastácio, Sara Cerejeira Fernandes, Regina Ferreira Alves, Celeste Meirinho Antão, Paula Oliveira Carvalho, Silvana Margarida Benevides Ferreira, Maria Isabel Cabrita Condessa

**Affiliations:** 1CIEC—Research Centre on Child Studies, University of Minho, Campus de Gualtar, 4710-057 Braga, Portugal; sarac.fernandes@ulsedv.min-saude.pt (S.C.F.); regina.alves@ie.uminho.pt (R.F.A.); maria.id.condessa@uac.pt (M.I.C.C.); 2Unidade Local de Saúde Entre Douro e Vouga, 4520-211 Santa Maria da Feira, Portugal; 3LiveWell—Research Centre for Active Life and Wellbeing, Polytechnic Institute of Bragança, Campus de Santa Apolónia, 5300-253 Bragança, Portugal; celeste@ipb.pt; 4Escola Superior de Saúde Norte Cruz Vermelha Portuguesa (ESSNorteCVP) Pedagogical Clinic, 3720-126 Oliveira de Azeméis, Portugal; paula.carvalho@essnortecvp.pt; 5Postgraduate Nursing Program, Faculty of Nursing, Federal University of Mato Grosso, Cuiabá 78060-900, Brazil; silvanabenevides3@gmail.com; 6Department of Education, Faculty of Human and Social Sciences, University of Azores, 9500-321 Ponta Delgada, Portugal

**Keywords:** SARS-CoV2, menstrual cycle, adolescents, vaccines

## Abstract

In a period globally known as long COVID, several post-acute infection sequelae and vaccination effects have been discussed. Objectives: This study aimed to identify the effects of COVID-19 infection and vaccines on the menstrual cycle of adolescents attending higher education and to verify the association between personal health factors and changes in their menstrual cycle after contact with the virus SARS-CoV-2 via infection or via the vaccine. Methods: A cross-sectional study was conducted using a questionnaire for data collection, applied online to Portuguese higher education adolescents aged between 18 and 24. The sample included 401 individuals. The statistical analysis of data was performed using SPSS. Results: More than half of the sample had a COVID-19 infection only once and took two doses of the vaccine. The mRNA Comirnaty 30 µg BioNTech vaccine was administered to 73.1%. The most common menstrual changes were an increase in blood clots, the blood becoming darker, shorter menstrual cycles, scarcer blood flow, and more irregular cycles. Menstrual changes correlated significantly with vaccination but not with infection. Conclusions: This study showed a lower percentage of women affected than other studies carried out closer to the pandemic period, which could mean that the effects are diminishing over time. Thus, adolescents’ menstrual health should be monitored.

## 1. Introduction

### 1.1. From Viruses’ Structure to SARS-CoV-2 Evolution

Viruses are pieces of genetic material made up of RNA or DNA genomes wrapped in a protein capsule that parasitise cells to replicate themselves. Their classification depends on the size, shape, chemical composition, and structure of the genome and the replication mode. Based on these, there are essentially 21 families of viruses, 14 RNA viruses and 7 DNA viruses [1]. Coronaviruses are positive-sense, single-stranded RNA viruses that infect humans and other animals. In humans, coronaviruses mainly cause infections of the superior respiratory tract. In other animals, in addition to respiratory infections, enteric, hepatic, and nervous system diseases have also been reported [2]. Like other RNA viruses, they have a high mutation rate and a high frequency of recombination, which facilitates adaptation to new hosts and ecological niches [2].

Some human coronaviruses cause severe acute respiratory syndrome (SARS), and until the epidemic of SARS-CoV in 2003, in China, only two human coronavirus genomes were completely available. This epidemic reinforced the study of coronaviruses and by December 2008, two more human coronaviruses were completely genome-sequenced (HCoV-NL63 and HCov-HKU1). After the discovery of these two new viruses, several patients in several countries with respiratory infections evidenced the presence of them [2].

A new and very similar human type of SARS-CoV emerged in the final weeks of 2019, also in China, giving origin to the COVID-19 pandemic [3,4].

The efficient infection and replication of SARS-CoV-2 has resulted in the emergence of new viral variants that have become predominant and raised concerns about their spread [3,4].

The evolution of coronaviruses is easily observable and measurable, like most RNA viruses. This evolution occurs on time scales comparable to events such as the transmission of the virus and its ecological dynamics (like changes in the number of infectious individuals over time, their immunity profiles, and human mobility). These evolutionary processes have a mutual impact that characterises the RNA virus [3,4].

This is determined by the rate at which mutations are generated and transmitted through populations. In this type of transmission, natural selection acts on an advantageous mutation (for example, the D614G mutation confers high transmissibility) [3,4,5]. The evolution of the virus involves additional complexity since replication and evolution take place within individuals [3].

The mutation rate of the virus is a determining factor for its evolution since it is where genetic changes arise per replication cycle (a biochemical property determined by the replication fidelity of a virus’s polymerase enzyme) [3].

The estimated mutation rate of SARS-CoV-2 is around 1 × 10^−6^–2 × 10^−6^ mutations per nucleotide per replication cycle. In addition to RNA replication errors, host-mediated genome alteration (innate cellular defence mechanisms) can substantially increase the number of mutations in the SARS-CoV-2 genome and thus influence its evolution [3,6].

During the Coronavirus Disease 2019 (COVID-19) pandemic, caused by SARS-CoV-2, prevention, diagnosis, and the treatment process were discussed on a global scale, and vaccine development studies were initiated. This research has produced positive results, and vaccines have been produced, ensuring protection against the SARS-CoV-2 virus and considerably reducing fatal complications [7].

### 1.2. SARS-CoV-2 and Its Influence on the Menstrual Cycle

It is well known that women’s health can be affected by a variety of factors, especially stress and infections. The COVID-19 pandemic has particularly affected the menstrual cycle, an important part of women’s lives. SARS-CoV-2 infection can interfere with the hypothalamus–pituitary–ovary–endometrium axis, causing changes in the menstrual cycle due to systemic inflammation caused by changes in the endometrium and vascularization [8].

The menstrual cycle is characterised by complex interactions between various tissues, hormones, and organ systems and can be altered by multiple physiological and pathological variables, including viral infections [9].

Most physiological processes in the female reproductive system involve inflammatory elements, namely cytokines and chemokines, which are regulators of the uterine environment. The inflammatory response interferes with the tissue repair, angiogenesis, degradation, remodelling, and proliferation of the endometrium [10].

COVID-19 is a pro-inflammatory disease which generates a cytokine storm and consequent immune exhaustion [11]. The effects of the SARS-CoV-2 virus on women’s health are accentuated by gender-related physiological differences [12].

After infection with SARS-CoV-2, many women report various changes in their menstrual cycle which are more frequent in those with greater symptoms caused by the infection [11,13,14], this being a sufficient stress factor to alter ovarian function [14].

Changes in the menstrual cycle can be related to the acute-phase immune response, inflammation, endogenous hormone levels, and/or immune cells in the endometrium [11,15]. 

In September 2020, a study of Chinese women examined how the SARS-CoV-2 infection affected menstruation, sex hormones, and ovarian reserves, revealing that 25% presented menstrual volume changes and 28% had menstrual cycle changes, with a decrease in volume in 20% of the women and a prolonged cycle in 19%. The average sex hormone and Anti-Müllerian hormone (AMH) concentrations of women of childbearing age with COVID-19 were not different from those of age-matched controls [13].

Some reports described the general effects of COVID-19, but evidence linking vaccination to menstrual cycle characteristics is limited. However, the alterations mentioned by different women imply an impact on their quality of life, so it is pertinent to delve into future studies focused on the subject [16].

Another study was conducted in 2021 in Turkey, with 586 women, to investigate the effects of COVID-19 vaccines on the menstrual cycle and showed that the most common menstrual changes after vaccination were delayed menstruation (30.0%) and a prolonged menstrual duration (22.5%). These two symptoms, as well as heavy bleeding and early menstruation, were significantly higher (*p* < 0.05) in women compared to before vaccination [17].

Another study conducted in the same year with 3958 American women aimed to determine the associations of SARSCoV-2 infection and vaccination against COVID-19 with the characteristics of the menstrual cycle. The results pointed to the non-existence of changes in the menstrual cycle for women infected with the SARS-CoV-2 virus and suggested that the COVID-19 vaccination may, in the short term, have altered the menstrual cycle, making it longer, specifically in women who presented variations in this cycle before the pandemic period. To achieve this analysis, behavioural variables related to stress obtained during the pandemic period were controlled [12].

Changes in the menstrual cycle following the administration of the first or second dose of the COVID-19 vaccination (regardless of the type of vaccine administered) have also been reported [12,16,17,18,19,20].

However, there seems to be a higher prevalence of menstrual cycle changes in women who experienced adverse effects from vaccination [18,19,20]. These irregularities were described as changes in the cycle length of less than five days and an increase in the amount of menstrual bleeding, changes which normalised in one to two subsequent cycles [11,15,16,17,18,19]. This rapid return to normal cycles suggests that the menstrual changes caused by COVID-19 may be a consequence of transient changes in sex hormones caused by the suppression of ovarian function, which resumes quickly after recovery [11,21].

Social stress related to the COVID-19 pandemic, independent of infection and vaccination, has also been reported as a potential disruptor of the menstrual cycle [7,21,22].

### 1.3. The Study’s Starting Point, Aims, and Hypothesis

Often, when discussing human physiology and health issues with our students in university classes, we realised that some of them reported experiencing different menstrual symptoms after infection and/or receiving the COVID-19 vaccine. This observation led us to search the literature and design the present study. Thus, this study aimed to identify the effects of COVID-19 infection and the vaccine on the menstrual cycle of Portuguese adolescents attending higher education to verify the association between individual factors and changes in the menstrual cycle after contact with the virus SARS-CoV-2 via infection or via the vaccine. The null hypotheses we formulated were: H_0.1_—contact with SARS-CoV-2 via infection did not influence the menstrual cycle of higher education adolescents; H_0.2_—contact with SARS-CoV-2 via the vaccine did not influence the menstrual cycle of higher education adolescents; and H_0.3_—individual factors (like age, smoking, blood groups, hormonal contraceptives, herpes, and others) do not influence the menstrual cycle and symptoms after contact with SARS-CoV-2.

## 2. Materials and Methods

To test the null hypothesis, a cross-sectional analytical descriptive study was carried out. It was a survey, following a quantitative approach. For data collection, a questionnaire was developed and validated specifically for this research.

The questionnaire included a section with sociodemographic, hormonal, and comorbidity-related questions, considered individual factors or independent variables. These variables were sex (because we considered the possibility of intersex and transexual people), age, the age of menarche, thyroid dysregulation, permanent medication, hormonal contraception, smoking, blood group, having any herpes virus, COVID-19 infection, and vaccination. For vaccine questions, we included the list of vaccines approved for Portugal by the National Authority for Medicines and Health Products (INFARMED) [23]. The next sections included questions related to the dependent variables, which were the perceived changes related to the menstrual cycle, namely variability in the blood abundance, the cycle length, and irregularities, as well as physical, emotional, psychological, and cognitive symptoms during the menstrual period after contact with SARS-CoV-2 via infection and/or via the vaccine. In this text, we focus only on the menstrual cycle changes and their relation with the individual factors.

The questionnaire was prepared to be applied online using Google Forms. The introduction to the questionnaire included the consent form, and only by providing informed consent could the participants continue to fill out the questionnaire. In addition, participants could accept or not, as well as having the freedom to withdraw at any time. They were informed and assured of the voluntary participation, anonymity, and the confidentiality of the data.

Ethical procedures were assured according to the Helsinki Declaration for research with human beings, and the study protocol and the questionnaire with informed consent were submitted to the Ethics Committee of the Instituto Politécnico de Bragança for approval in April 2024. We received the first communication in the final week of May and the final approval was obtained at the beginning of July (protocol code: 522348; 5 July 2024).

### 2.1. Participants and Sampling Process

We tried to obtain a representative sample of the elected population: adolescents aged between 18 and 24 years, attending higher education (universities and polytechnic institutes) in continental Portugal and the Portuguese Autonomous Regions of the Azores and Madeira islands. For the academic year of 2023/24, there were 233,649 women in Portuguese higher education [24]; some of them were more than 24 years old. Nevertheless, based on this total population, the sample size was calculated considering a confidence interval of 95% and an error of 5%. The final sample size estimated was 384 individuals [25]. Nevertheless, we cannot consider it as a representative sample because it was not an aleatory and probabilistic sampling process.

The researchers worked on the dissemination of the questionnaire, starting with their students and following this with colleagues at other universities and polytechnic institutes and known people with familial relationships with girls fulfilling the inclusion criteria. Contacts were established with all the Portuguese public universities and polytechnic institutes, as well as with Portuguese organisations of university students. In this way, it was a snowballing and respondent-driven sample [25] of the whole country. In terms of inclusion criteria, we defined two: to be aged between 18 and 24, because 18 is the lowest age of first-year higher education students in the second semester and 24 is considered the age of the end of adolescence [26]; and to have had contact with SARS-CoV-2 via infection and/or via the vaccine.

We obtained a total of 433 answers. Nevertheless, 28 participants were more than 24 years old and were immediately rejected. Analysing the sample sociodemographics, it was verified that 4 adolescents had had neither COVID-19 disease nor a vaccine, and they were also rejected. Thus, the final sample consisted of 401 individuals.

### 2.2. Data Analysis and Reliability

Data obtained from the questionnaire were automatically registered in an Excel file, which was imported to the IBM Statistical Package for Social Sciences (SPSS) (version 29.0) software. A descriptive analysis was performed to characterise the sample and determine the frequency of the dependent variables. Afterwards, statistical tests were applied to observe the central tendency measures and associations between variables. Student’s T-test was applied between the dependent variables of menstrual cycle changes and the individual factors (or independent variables): the contraceptive pill to regulate the menstrual period, a thyroid disorder, herpes, smoking, and the gestation length. Nonparametric Spearman correlations were tested for between the menstrual cycle changes and blood groups, the menstrual period’s regularity, the menstrual period length, the number of COVID-19 infections, and the number of vaccine doses. Correlations were classified as small (0.10), medium (0.30), or large (0.50) [25]. To validate the internal consistency of the question related to the menstrual cycle changes, the reliability Cronbach Alpha (α) test was applied and strong values (>0.90) were obtained for the variable menstrual cycle changes, with 10 items and a Likert scale with four scores from “totally true = 1” to “totally false = 4”, α = 0.96. This value shows very highly reliability [25], enabling us to be confident in this questionnaire carried out based on the changes referred to by adolescents in classes and by some of the literature [27,28,29]. The significant *p*-value for all the tests was established at *p* < 0.050 and the confidence interval (CI) at 95%.

## 3. Results

### 3.1. Sociodemographics, Hormonal Factors, and Comorbidities

The sample included 401 adolescents aged between 18 and 24 years (M = 20.73 ± SD = 1.76), with 99.5% declaring themselves to be females and 0.5% preferring not to say. Regarding gender, 92.3% said they considered themselves women/girls/very feminine, while 6.0% revealed themselves to have a tendency towards activities and characteristics more associated with men, such as a “mary boy”, 0.7% identified themselves as non-binary, and 1.0% chose not to say. Regarding sexual orientation, 88.0% said they were heterosexual, followed by 9.2% who said they were bisexual, 1.7% who said they were lesbian, and 1.0% who preferred not to say. The majority were undergraduate students (Table 1).

Focusing on the adolescents’ health profiles, regarding endocrine function, 6.2% had a thyroid disorder, and 48.9% (n = 196) said they took medication regularly. Nevertheless, the most frequent medicines referred to were contraceptive pills (n = 117; 29.2%), followed by antidepressants (2.0%) and thyroxine (1.5%). Several combinations were found, namely contraceptives and antidepressants (1.2%), thyroxine and contraceptives (0.7%), and others varying between 1.0% and 0.5%. For the specific question about pills to regulate the menstrual cycle, nearly two-thirds of the sample (66.3%) answered yes. Only 16.5% of the sample were smokers. Regarding the blood group, 179 participants (44.6%) did not know their group, and the most frequent was group O (n = 99; 24.7%), with predominant O+ (n = 71; 17.7%), while O− represented only 6.2% (n = 25). Found in a similar proportion was group A (n = 94; 23.4%), with the majority of these being A+ (n = 75; 18.7%). Group B represented 4.7% of the total sample (n = 19), and the smallest proportion was group AB (n = 10; 2.5%). Considering other viruses, 16.0% indicated that they had herpes. Regarding birth, 14.2% were born prematurely. The average menarche was 12.32 (SD = 1.53), with a minimum of 8 and a maximum of 18 years old. The modal age of menarche was 12, registered by nearly a third (32.9%) of the participants (Table 2).

Regarding the menstrual cycle’s regularity, more than half (55.0%) said they had a regular cycle, while only 12.0% considered it irregular (Figure 1a). The length of the menstrual period was 3–5 days for the majority (60.0%) and 5–7 for 35.0%. Few cases of more than 7 days (2.0%) and less than 3 days (3.0%) were registered (Figure 1b).

### 3.2. Contact with SARS-CoV2 Virus via Infection and Vaccination

More than half of the sample had had COVID-19 only once, while 26% had never had the disease (Figure 2a). Regarding vaccination, the majority (56%) had received two doses of the vaccine, and 29% took three doses. Only one person was not vaccinated, and the maximum number of doses received was four (Figure 2b).

Regarding the type of vaccine adolescents took, Table 3 shows a diverse list of either only one type or a combination of different vaccines. Nevertheless, the Comirnaty 30 µg mRNA vaccine was the most applied to this sample (73.1%).

Given the diversity, these vaccination alternatives were grouped, considering (Table 3) (a) the group that took only mRNA vaccine(s); (b) the group that received only adenovirus vaccine(s); (c) and the group that received a combination of mRNA and adenovirus vaccines. The results showed that a great majority only received mRNA vaccines (n = 331; 82.5%), few received adenovirus vaccines (n = 9; 2.2%), and a little more were injected with a combination of mRNA and adenovirus vaccines (n = 31; 7.7%).

### 3.3. Menstrual Cycle Changes

An increase in blood clots was the most referred-to change in the menstrual cycle, mentioned by 11.5% of the adolescents. The second most noted change was that the colour of the blood became darker, referred to by 10.2%. Also, nearly 8% of adolescents mentioned shorter menstrual cycles (8.4%), scarcer blood flow (7.7%), and more irregular cycles (7.9%). Some variables evidenced a significant influence on some menstrual cycle changes. More precisely, adolescents with a thyroid disorder perceived that their cycle became scarcer more compared to those without this condition. The use of contraceptive pills revealed some influence because those who did not use them perceived that their cycles became irregular significantly more. The smokers noted more than the non-smokers that the colour of their menstrual blood became darker, as did those of premature birth. These four factors revealed a significant influence (*p* < 0.050) on the menstrual cycle instead of a little effect as measured by Cohen’s *d* (Table 4).

Other factors were tested, namely age, gender, sexual orientation, grade, blood group, regular medication, and usually having herpes, which did not show significant differences.

Regarding correlations, the age of menarche did not correlate with changes in the menstrual cycle after COVID-19 infection or vaccination. On the other hand, the menstrual period’s regularity and length correlated significantly, although weakly, and positively with the changes in the menstrual cycle (Table 5). Concerning regularity, only the item about the cycle becoming irregular showed a significant and positive correlation, which means that those who considered their cycle more regular also considered the item related to their cycle becoming irregular falser. The length of the menstrual period also correlated significantly and positively with the cycle becoming shorter, the blood flow scarcer, and the cycle more irregular and the blood colour becoming lighter. This means that the longer the menstrual period, the falser these changes were.

Correlations between the number of times they had had COVID-19 and the number of vaccine doses were also tested for. Significant correlations were only found between the menstrual cycle changes and the vaccine. Four items evidenced weak, negative, but significant correlations (Table 6), which means that the more vaccine doses, the truer these statements were (the cycle being shorter and more regular, the blood scarcer and lighter).

Apart from the number of doses, the diversity and type of vaccine were also correlated with the menstrual cycle changes. Firstly, we grouped data to test differences between those who always took the same vaccine in several shots, and those who took different vaccines at different points. Being two groups, Student’s T-test was applied. The results indicated significant differences for two items: the blood became scarcer, and the clots increased. Adolescents who received several vaccines reported more of these symptoms than those who always received the same vaccine, although Cohen’s d values were small (Table 7).

Then, data were tested regarding the type of vaccine (mRNA, adenovirus, and mRNA + adenovirus), as presented above in Table 3, categorised into three groups (a, b, and c, respectively). The nonparametric Kruskal–Wallis test only detected significant differences for the topic “the clots decreased” (*H* = 6.436; *p* = 0.040). Searching for the differences between pairs of groups, the Mann–Whitney test found a significant difference between those who received only mRNA vaccines and those who received only adenovirus vaccines (z = −2.070; *p* = 0.038), suggesting that the statements were falser for those who received mRNA vaccines.

## 4. Discussion

This study analysed the impact of COVID-19 vaccination and SARS-CoV-2 infection on the menstrual cycle of adolescents attending Portuguese higher education and the influence of individual factors on these changes. Overall, around 10 per cent of teenagers experienced changes in their menstrual cycle, the most noticeable being the formation of clots, followed by the blood becoming darker. More than half of the sample had the infection only once and took two doses of the vaccine. The great majority took the Comirnaty 30 µg BioNTech mRNA vaccine. Our study evidenced a lower percentage of women affected than the study carried out in 2020 with Chinese women infected by SARS-CoV-2 [13] and the study conducted in Turkey in 2021 related to COVID-19 vaccination effects [17]. This could mean that the effects are diminishing over time, or our lower percentages can be due to memory effects.

The results revealed no significant correlations between the number of COVID-19 infections and changes in the menstrual cycle. However, changes in the menstrual cycle demonstrated a significant positive correlation with the number of vaccine doses, associated with shorter and more regular cycles and reduced menstrual flow. Moreover, the diversity and type of vaccine influenced the changes in the menstrual cycle. The adolescents administered with different vaccines exhibited a greater incidence of symptoms, including diminished blood flow and heightened clot formation, in contrast to those who received identical vaccines throughout all doses. While a Spanish [12] study highlighted significant differences between those who received mRNA vaccines and those who received adenovirus vaccines, our study directs more attention to those who received a mix of vaccines. Nevertheless, the number of cases we had was low, which is claimed to deepen differences between studies.

Our first null hypothesis (H_0.1_—contact with SARS-CoV-2 via infection did not influence the menstrual cycle) was accepted. This question has been investigated in multiple research studies, indicating a complex relationship between the virus and changes in the menstrual cycle. Some studies suggest that COVID-19 disease can lead to menstrual changes, such as delays, shorter menstrual flows, and a decreased flow volume [30], with more severe cases experiencing greater monthly changes [30,31,32,33]. Other studies indicate that these changes are not significant or long-lasting, like the results of the present study. For example, a study conducted in Dubai indicated that 90% of participants did not experience dysmenorrhoea, and 81% reported no change in the menstrual flow rate [34]. In a study of 1,335 women in the United Kingdom, 56% of women reported that they had experienced menstrual cycle changes and dysmenorrhoea during the pandemic [35]. Although SARS-CoV-2 can temporarily impact menstrual cycles, these changes are often not enduring. For example, in the study by Al-Bdairi and colleagues [33], most women reported a return to their pre-infectious menstrual patterns within one or two months after being infected.

The second hypothesis proposed (H_0.2_—contact with SARS-CoV-2 via the vaccine did not influence the menstrual cycle) was rejected. This is consistent with the existing literature. A recent literature review [36] showed changes in the duration of the menstrual cycle, flow, and menstrual pain after COVID-19 vaccination. Other studies conducted in different countries also support this hypothesis. For example, in research conducted among Saudi women, 65.6% reported irregularity in their menstrual cycle (timing, blood flow, and pain) after receiving the COVID-19 vaccine [37]. These symptoms happened after the first and second doses and were similar regardless of the type of vaccine. Similarly, a study in Iraq showed that 12.2% and 13.7% of women experienced significant menstrual changes after the first and second doses of the Pfizer vaccine, respectively. These changes included irregularities in the length of the cycle and changes in the quantity and duration of menses [38]. In Italy, Granese and colleagues [39] found a shift from regular to irregular menstrual rhythms and changes in the menstrual duration and flow volume post-vaccination. In Jordan, 40% of participants experienced changes in the menstrual duration, and around 20% reported heavier bleeding and more severe premenstrual syndrome symptoms after vaccination [40]. However, these effects were short-term, with most women reporting a return to normal menstrual patterns within three to six months [37,39].

Our study presents results similar to those of the American one [12] where infection did not imply changes in the menstrual cycle, but vaccination did.

For the last hypothesis formulated (H_0.3_—individual factors (like age, smoking, blood groups, hormonal contraceptives, herpes, and others) do not influence the menstrual cycle changes after COVID-19 infection and/or vaccination), we can partially accept it.

The analysis of individual factors showed that not all of them affected the menstrual cycle of adolescents after COVID-19 infection or vaccination. Only smoking, hormonal contraceptives, a thyroid disorder, and pre-term birth were associated with a higher likelihood of reporting menstrual changes after COVID-19 infection or vaccination. In our study, adolescent smokers had a higher score for the item ‘The colour of the blood became darker’ compared to non-smoking adolescents. According to research conducted by Alvergne and colleagues, it appears that smoking is a strong predictor of post-vaccination menstrual changes (smokers were 44% more likely to have menstrual irregularities than non-smokers) [41]. Our study also shows that hormonal contraceptives may help to reduce menstrual cycle irregularity caused by COVID-19 infection or vaccination. Similarly, other studies observed that women using combined oral contraceptives were less likely to have menstrual changes following vaccination than those not using such contraceptives [41,42]. Our study found that adolescents with thyroid disorders were more likely to experience the blood flow becoming scarcer compared to those without these conditions, and those of premature birth noted that the colour of their menstrual blood became darker more. However, we have not found any specific studies addressing the influence of thyroid disorders and premature birth on these changes. In this sense, our study brings new scientific knowledge and drives us to deepen our understanding of these topics.

Although innovative knowledge is provided, we recognise some limitations of this study. The type of sampling could be a limitation because it was not a probabilistic sample. Nevertheless, to minimise this limitation, we increased the number of participants beyond the sample size statistically recommended and asked for the application of the questionnaire in all Portuguese higher education institutions, trying to cover the whole country. Another limitation could be the memory effect, since three years passed from the beginning of vaccination to the data collection. Nevertheless, it was important for us to know the long-term effect of COVID-19 infection and vaccines on adolescents’ menstrual cycles, reminding us that these teens were aged between 15 and 21 when they were exposed to SARS-CoV-2. The lower number of cases vaccinated with adenovirus vaccines and a mix of mRNA and adenovirus vaccines is another limitation which requires deeper research into the type of vaccine.

## 5. Conclusions

This research characterises Portuguese adolescents aged 18–24 attending higher education and exposed to SARS-CoV-2 and highlights the importance of monitoring menstrual health in women vaccinated against COVID-19. As the key findings, our results revealed the following: the most common changes were an increase in blood clots, blood becoming darker, shorter menstrual cycles, scarcer blood flow, and more irregular cycles; there were significant associations between changes in menstrual cycles and the COVID-19 vaccine, but not SARS-CoV-2 virus infection; the use of different vaccines seemed to imply more changes than the use of only one type; and menstrual cycle changes affected a reduced percentage of adolescents and tended to be temporary. 

As practical implications, our results may help to ease some fears and encourage more people to get vaccinated, to promote adolescents’ understanding of other effects like those of smoking and thyroid disorders, and to inform health authorities about a better choice of COVID-19 vaccines for adolescents.

Given the results, more research is needed to better clarify the effects of vaccines on menstrual health. As future research, we recommend regular data collection focusing on a more diverse and representative sample of the Portuguese population. This should include women with different demographic profiles to study any changes in menstrual cycles before and after vaccination and infection with this type of virus. Younger adolescents will be our next focus. In addition, we propose to conduct case–control studies to verify any causal relationship between COVID-19 vaccination and menstrual disorders. Future and longitudinal studies should also examine the effects on fertility and general reproductive health.

Furthermore, other variables, such as stress, anxiety, and depression, should be considered, since other studies [7,21,22] have found a relationship between stress and menstrual cycle changes. This emphasises the need for mental health support, especially for women with menstrual irregularities.

In this sense, healthcare providers should be aware of the potential impact of the COVID-19 vaccination on menstrual cycles and be prepared to offer appropriate guidance and interventions to adolescents who experience these changes. In the same way, health educators have an important role in guiding adolescents to good menstrual health and to avoiding vaccination hesitancy, given the benefits are higher than the risks.

## Figures and Tables

**Figure 1 healthcare-13-00002-f001:**
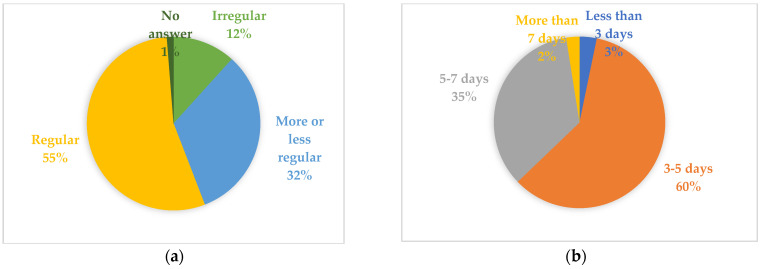
(**a**) Regularity of the menstrual cycle; (**b**) menstrual cycle length.

**Figure 2 healthcare-13-00002-f002:**
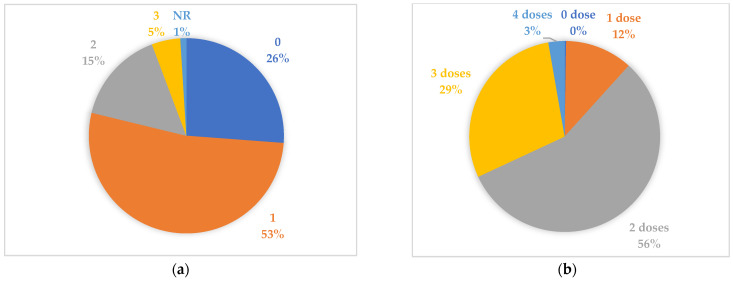
(**a**) Number of times having COVID-19 disease (NR = no response); (**b**) number of vaccination doses.

**Table 1 healthcare-13-00002-t001:** Sociodemographics of the sample.

Variable	Minimum	Maximum	Mean (SD)
Age	18	24	20.73 (1.76)
	Categories	n	%
Sex	Female	399	99.5%
	Intersex	0	0.0%
	Prefer not to say	2	0.5%
Gender	Woman/Girl/Very feminine	370	92.3%
	With a tendency towards activities and characteristics more associated with boys	24	6.0%
	Non-binary	3	0.7%
	Prefer not to say	4	1.0%
Sexual orientation	Heterosexual	353	88.0%
	Homosexual/Lesbian	7	1.7%
	Bisexual	37	9.2%
	Prefer not to say	4	1.0%
Higher education	1st-year undergraduate	109	27.2%
	2nd-year undergraduate	80	20.0%
	3rd-year undergraduate	125	31.2%
	1st year of master’s programme	59	14.7%
	2nd year of master’s programme	25	6.2%
	1st year of doctorate	3	0.7%

**Table 2 healthcare-13-00002-t002:** Health characteristics of the sample.

Variable	Yes		No	
	n	%	n	%
Thyroid disorder	25	6.2%	376	93.8%
Regular medication	196	48.9%	205	51.1%
Smoker	66	16.5%	335	83.5%
Herpes	64	16.0%	337	84.0%
Premature birth	57	14.2%	338	84.3%
Pill to regulate the menstrual cycle	266	66.3%	135	33.7%
Age of menarche	Minimum	Maximum	M (SD)	Mode
	08	18	12.32 (1.53)	12

**Table 3 healthcare-13-00002-t003:** Type of vaccine administered.

Vaccine(s)	n	%
Pfizer (Comirnaty 30 µg, BioNTech) ^(a)^	293	73.1%
Pfizer (Comirnaty 30 µg, BioNTech) + AstraZeneca (Vaxzevria) ^(c)^	19	4.7%
Moderna (Spikevax) ^(a)^	17	4.2%
Moderna (Spikevax) + Pfizer (Comirnaty 30 µg, BioNTech) ^(a)^	13	3.2%
AstraZeneca (Vaxzevria) ^(b)^	7	1.7%
Comirnaty bivalent Original/Omicron ^(a)^	5	1.2%
Moderna (Spikevax) + AstraZeneca (Vaxzevria) ^(c)^	4	1.0%
Pfizer (Comirnaty 30 µg, BioNTech) + Jansen (Jcovden) ^(c)^	3	0.7%
Jansen (Jcovden) ^(b)^	2	0.5%
Pfizer (Comirnaty 30 µg, BioNTech) + AstraZeneca (Vaxzevria) + Comirnaty bivalent Original/Omicron ^(c)^	2	0.5%
Moderna (Spikevax) + Pfizer (Comirnaty 30 µg, BioNTech) + AstraZeneca (Vaxzevria) ^(c)^	2	0.5%
Spikevax bivalent Original/Omicron ^(a)^	1	0.2%
AstraZeneca (Vaxzevria) + Comirnaty bivalent Original/Omicron ^(c)^	1	0.2%
Pfizer (Comirnaty 30 µg, BioNTech) + Spikevax bivalent Original/Omicron ^(b)^	1	0.2%
Pfizer (Comirnaty 30 µg, BioNTech) + Comirnaty bivalent Original/Omicron ^(a)^	1	0.2%
No answer	30	7.5%

^(a)^ mRNA vaccines; ^(b)^ adenovirus vaccines; ^(c)^ mRNA + adenovirus vaccines.

**Table 4 healthcare-13-00002-t004:** Menstrual cycle changes and influencing factors (Student’s T-test).

	n	M (SD)	Factor	*t*	*p*-Value	*d*
The menstrual cycle became shorter.	398	3.35 (0.69)				
The menstrual cycle became longer.	397	3.42 (0.61)				
The blood flow became more abundant.	398	3.37 (0.65)				
The blood flow became scarcer.	398	3.35 (0.67)	Thyroid disorder	−2.095	0.037	−0.433
The cycle became more irregular.	399	3.35 (0.66)	Contraceptive pill	2.055	0.041	0.232
The cycle became more regular.	398	3.43 (0.58)				
The color of the blood became darker.	398	3.31 (0.73)	Smoker	−2.087	0.038	−0.281
	398	3.31 (0.73)	Pre-term birth	−2.532	0.012	−0.363
The color of the blood became lighter.	398	3.44 (0.55)				
The clots increased.	399	3.29 (0.75)				
The clots decreased.	399	3.44 (0.57)				

(Likert scale: 1 = totally true; 2 = true; 3 = false; 4 = totally false).

**Table 5 healthcare-13-00002-t005:** Menstrual cycle changes and menstrual period’s regularity and length (Spearman’s correlation).

	Correlation’s Parameters	Regularity	Length
The menstrual cycle became shorter	*ρ*	0.055	0.154 **
	*p*	0.275	0.002
	N	393	398
The blood flow became scarcer	*ρ*	0.085	0.127 *
	*p*	0.094	0.011
	n	393	398
The cycle became more irregular	*ρ*	0.161 **	0.108 *
	*p*	0.001	0.031
	n	394	399
The color of the blood became lighter	*ρ*	0.069	0.113 *
	*p*	0.171	0.025
	n	393	398

* Significant at the level of *p* < 0.050; ** significant at the level of *p* < 0.010.

**Table 6 healthcare-13-00002-t006:** Menstrual cycle changes and contact with SARS-CoV-2 (Spearman’s correlation).

	Correlation’s Parameters	Number of Times Having COVID-19	Number of Vaccine Doses
The menstrual cycle became shorter	*ρ*	0.027	−0.123 *
	*p*	0.597	0.014
	n	394	398
The blood flow became scarcer	*ρ*	0.043	−0.121 *
	*p*	0.398	0.015
	n	394	398
The cycle became more regular	*ρ*	0.050	−0.104 *
	*p*	0.324	0.037
	n	394	398
The color of the blood became lighter	*ρ*	0.007	−0.130 **
	*p*	0.882	0.009
	n	394	398

* Significant at the level of *p* < 0.050; ** significant at the level of *p* < 0.010.

**Table 7 healthcare-13-00002-t007:** Menstrual cycle changes and vaccine diversity (Student’s T-test).

	Vaccines	n	M (SD)	*t*	*p*-Value	*d*
The blood flow became scarcer	one	324	3.37 (0.64)	2.317	0.021	0.365
	several	46	3.13 (0.81)			
The clots increased	one	325	3.31 (0.72)	2.654	0.008	0.418
	several	46	3.00 (0.94)			

## Data Availability

The data presented in this study are available on request from the corresponding author due to privacy and ethical reasons.

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
