# Peer review of "Relation Between COVID-19 Infection and Vaccine and Menstrual Cycle Changes of Portuguese Adolescents in Higher Education"

_healthcare, 2024, doi:10.3390/healthcare13010002_

Round 1

Reviewer 1 Report

Comments and Suggestions for Authors

Thank you for allowing me to review this study, good effort on a very interesting and much needed study. These are my comments:

The introduction is very long, it can be divided into subheadings. 

Line 427-428:  Firstly, the type of sampling because despite data collection covering the whole 427 country, it was not a probabilistic sample. This sentence needs to be rewritten to be more clear. 

Line 123: typo

Methodology section
  1. While the authors mention that the questionnaire was reviewed by two experienced researchers and piloted, further details on the validation process  like using Cronbach’s alpha for reliability would strengthen the study's credibility.
  2. The manuscript does not clarify whether the sample is representative of the broader population of Portuguese young adults in higher education. Addressing potential biases in sample selection (e.g., socioeconomic status, accessibility to technology) would enhance the generalizability of the findings.
  3. Authors mentioned some limitations in the conclusion indicated they missed some important variables that could influence menstrual health like stress and depression, however additional variables were also missing like dietary habits and lifestyle changes. 
Conclusion section

  1. Conclusion can be structured more clearly by dividing into subsections such as "Key Findings, Future Research, and "Practical Implications."
  2. Briefly discuss how the findings align with global research trends or could inform vaccination policies.
  3. Authors can elaborate on why vaccination, but not infection, was correlated with menstrual changes, particularly in light of earlier studies suggesting both factors might play a role.

Author Response

Dear Reviewer,

We thank you very much your time and comments to our manuscript. Please see our detailed responses in attachment.

Best wishes,

Reviewer 2 Report

Comments and Suggestions for Authors

Dear authors, this manuscript is about an issue that has to be investigated in more detail.

In my opinion in the introduction there are more details about general information about COVID-19, you should focuse on the menstrual cicle and Sars cov 2 infection.

In the section Material and Methods the study design is not clearly presented. I do not understand which is the control group.

Also in Table 2, many of the participanys took contraceptive pills, which represent an important bias, because we know that those pills have an important role in controlling the menstrual flux and pain.

I think it would be better to compare the adolescents who were vaccinated, or infected with Sars Cov 2, but under 18 years old, who did not took contraceptive pills with a similar group in a similar period of time before the  pandemic.

Good luck and thank you.

Author Response

Dear Reviewer,

Many thanks for your constructive comments. We hope to have corresponded to all the suggestions you gave us.

Please see detailed responses in the attachment. 

Reviewer 3 Report

Comments and Suggestions for Authors

The manuscript addresses an under-researched topic: menstrual changes in adolescents after COVID-19 infection and vaccination. It provides new data on Portuguese adolescents, an underrepresented demographic in similar studies. However, the novelty is somewhat limited due to the related studies globally.

The objectives are clear, focusing on the effects of COVID-19 infection and vaccination on menstrual health and identifying influencing personal factors.

The hypotheses are well-formulated but could be more concise. Statistical methods are correctly applied.

The study highlights temporary menstrual changes and their correlation with vaccination, but it could discuss more about the implications for healthcare practices or vaccine hesitancy.

The writing is clear but could be shortened in some sections, particularly in the introduction, discussion and conclusion. The manuscript follows a logical structure.

The Figures 1,2 are not well visible, especially the yellow color parts. Consider use B&W with shading or patterns.

Line 31-41: Consider simplifying or remove the description of virus classification, since it may be distracting from the focus.

Line 77-142: The connection between the menstrual cycle and COVID-19 is strong but could be stated more concisely.

Line 162-163: Including sex variability is very interesting point. Could you please add some background information or more discussion about this point.

Line 167-172: Include a brief explanation for why only these variables were considered? Please cite if the variables selected based on previous studies.

Lines 185-190: could be removed as it overlapped with Lines 482-484.

Line 219: Student’s T-test.

Line 359: "This could mean that the effects are diminishing over time" should include possible alternative explanations, such as memory effect.

The Conclusion could be shortened, emphasizing the key findings and the practical implications for healthcare and vaccination.

Author Response

Dear reviewer,

Many thanks for your comments to our manuscript and for the time you spent with it. We acknowledge all your suggestions. Please, see the attachment for detailed answers and explanations.

Best regards,

Round 2

Reviewer 2 Report

Comments and Suggestions for Authors

Dear authors, the manuscript looks better. Also the additional information regarding the study design are helpful.